# Optimizing Ultrasonic Welding Parameters for Multilayer Lap Joints of PEEK and Carbon Fibers by Neural Network Simulation

**DOI:** 10.3390/ma15196939

**Published:** 2022-10-06

**Authors:** Sergey V. Panin, Dmitry Yu. Stepanov, Anton V. Byakov

**Affiliations:** 1Department of Innovative Programs and Projects, National Research Tomsk State University, 634050 Tomsk, Russia; 2Laboratory of Mechanics of Polymer Composite Materials, Institute of Strength Physics and Materials Science SB RAS, 634055 Tomsk, Russia

**Keywords:** machine learning, neural network simulation, ultrasonic welding, lap joint, PEEK, prepreg, interface, adhesion, Taguchi method

## Abstract

The aim of this study is to substantiate the use machine learning methods to optimize a combination of ultrasonic welding (USW) parameters for manufacturing of multilayer lap joints consisting of two outer PEEK layers, a middle prepreg of unidirectional carbon fibers (CFs), and two energy directors (EDs) between them. As a result, a mathematical problem associated with determining the optimal combination of technological parameters was formulated for the formation of USW joints possessing improved functional properties. In addition, a methodology was proposed to analyze the mechanical properties of USW joints based on neural network simulation (NNS). Experiments were performed, and threshold values of the optimality conditions for the USW parameters were chosen. Accordingly, NNS was carried out to determine the parameter ranges, showing that the developed optimality condition was insufficient and required correction, taking into account other significant structural characteristics of the formed USW joints. The NNS study enabled specification of an extra area of USW parameters that were not previously considered optimal when designing the experiment. The NNS-predicted USW mode (*P* = 1.5 atm, *t* = 800 ms, and *τ* = 1500 ms) ensured formation of a lap joint with the required mechanical and structural properties (*σ*_UTS_ = 80.5 MPa, *ε* = 4.2 mm, *A* = 273 N·m, and Δ*h* = 0.30 mm).

## 1. Introduction

Ultrasonic welding (USW) is a common method for joining thermoplastic materials (PE, PP, PA, etc.) and their composites, primarily for structural applications. This technology and equipment have been well-developed and widely used in various industries. The USW process includes three main stages:

- Initially, the clamping force is gradually increased to squeeze the parts being welded. At this stage, ultrasonic (US) vibrations are not applied.

- Then, US vibrations of a given frequency and amplitude are initiated at the same clamping force. In this case, frictional heating and melting of the mated surface layers of the welded parts occurs due to their mutual movement relative to each other. As a result, processes of mixing, mass transfer, and mutual penetration of polymer chains develop.

- The third stage involves molten material solidification at the same clamping force. US vibrations are no longer applied.

The above-mentioned USW parameters (clamping force and durations of both application of US vibrations and holding under pressure after the end of the US application) can exert opposite (competing) effects on the structure formation. Additionally, it is necessary to take into account some properties of the joined polymers, such as the glass transition and melting temperatures, their molecular structures (molecular weight and polarity), etc. It is known that the diffusion rheological interaction of molecules typically occurs at the USW joint interface with the formation of a structural transition zone under pressure. Such mutual penetration develops when the polymers are in a viscous state and their molecules possess the maximum mobility and the minimum packing density [1,2,3,4,5].

The USW method has been applied in high-tech industries to join continuously reinforced polymer composites based on high-performance polymers (HPP). In [6,7,8,9], it was demonstrated that a polymer film (energy director (ED)) should be placed between the parts to be joined by USW. A reliable USW joint can be formed as a result of ED melting. If the lap-joint shear strength (LSS) is comparable to that of a polymer binder, the USW parameters are considered to be optimal. The subject of research in such a study is the optimization of the USW parameters, which, among other things, depend primarily on the material and structure of the ED [10,11,12,13].

Many authors have discussed and used various methods to optimize technological parameters, including contradicting parameters [14]. Multifactorial experiments have been designed with a large number of “control” parameters, in particular, applying the Taguchi method [15], which enables ranking of the parameters according to their degree of influence. Then, their “best” levels are selected from a set of measured values, i.e., optimal or as close as possible to optimal. The key drawback of such studies is the high costs (both time and financial) of the experiments. If the experimental data sample is small, experimental parameters can be set outside the optimal range. Therefore, it is beneficial to use mathematical methods for computer simulation of investigated processes, together with experimental design methods, to predict functional properties of multilayer lap joints that can be formed with arbitrary USW parameter values. This requires the involvement of up-to-date approaches, including the machine learning methods [16,17,18,19].

Lap shear strength (LSS) [20] is a conventionally estimated strength parameter for lap joints of laminates welded using ultrasonic vibrations. The ASTM D5686 has been employed for adhesively bonded joints of polymer composites [21]. A complete list of mechanical tests employed for polymer composite welds includes double-lap shear [22], interlaminar shear strength (ILSS) [23], cross tensile [24], pull through [25], three-point bending [26], double cantilever beam, and end-notched flexure [1].

In addition to continuous fiber or fabric reinforced polymer laminates, US welding is applied to join particulate polymer composites, the strength properties of which are much lower than those of reinforced polymer laminates [27,28,29,30,31,32]. Regardless of the particular type of adherends, an increased LSS can be attained by embedding reinforcing fabric between the components to be welded. Because it is difficult to achieve US welding of a polymer with a fabric, prepreg can be used for interface reinforcement. However, the selection of particular prepreg parameters is a complex problem. Furthermore, USW technology can be applied as a repair technique when a part of damaged composites is removed with subsequent “patch” installation. This approach is widely used for damage repair of epoxy composites [32], defining the problem under investigation in the present study.

The aim of this study was to substantiate the use of machine learning methods to optimize a combination of USW parameters for manufacturing of multilayer lap joints consisting of two outer PEEK layers, a middle prepreg of unidirectional carbon fibers (CFs), and two EDs between them.

## 2. Problem Statement

Objectives of the study included the development of a methodology for determining the optimal conditions (combination of USW parameters), providing required mechanical and structural properties of multilayer lap joints based on the results of a limited number of experimental investigations. In the general case, the USW parameters are: (a) frequency of US vibrations (*ω*), (b) the sonotrode vibration amplitude (*θ*), (c) duration of US vibrations (*t*), (d) clamping duration (of the sonotrode) after the end of US vibrations (*τ*), clamping pressure (*P*) (of the sonotrode); (f) porosity of the ED ($\nu_{ED}$), (g) polymer/fiber content in the ED ($\varphi_{ED}$), and (h) the thickness of the ED ($h_{ED}$). This set of specified USW parameters was designated as their vector, $\vec{\rho }={\left(\omega ,\theta ,t,\tau ,P,\nu_{ED},\varphi_{ED},h_{ED}\right)}^{T}$. In this paper, it is assumed that values of a number of the parameters, i.e., *t*, *τ*, and *P* can be varied (to optimize the USW process) within preset ranges. The other parameters (*ω*, *θ*,$\nu_{ED},\varphi_{ED},h_{ED}$) are constant due to the USW machine specification. The type and dimension of the ED and prepreg were also held constant.

The following USW joint properties were used as the key mechanical characteristics: (a) ultimate tensile strength ($\sigma_{UTS}$), (b) elongation at break (*ε*), (c) work of strain (*A*) (area under the stress–strain curve of the corresponding diagram); and the following structural parameter was applied: (d) USW joint thinning (during USW) (Δ*h*). These properties were designated through a vector, $\vec{R}={\left(\sigma_{UTS},\varepsilon ,A,\Delta h\right)}^{T}$, the experimentally measured values of which can vary randomly to a certain extent due to the heterogeneous (multidirectional) influence of the USW parameters (for example, the clamping pressure (*P*) and duration of US vibrations (*t*)). In addition to the influence of an arbitrary combination of USW parameters in the experiment, the pattern of this randomness also applies to “measuring” (metrological) factors: the error of measuring instruments, the locality of measurements (for example, the USW joint thickness), etc.

The optimal USW conditions are the values of the control parameters under which the USW joint properties satisfy the following system of inequalities:(1)$$\{\begin{array}{c} \sigma_{\min}<\sigma_{UTS}<\sigma_{\max},\\ \varepsilon_{\min}<\varepsilon <\varepsilon_{\max},\\ A_{\min}<A<A_{\max},\\ \Delta h_{\min}<\Delta h<\Delta h_{\max}, \end{array}$$
where the threshold values of the inequalities are the levels of both mechanical and structural properties of a material that satisfy its specifications (or are within an “allowed” range of the values).

Experimental data were obtained by conducting *N* physical experiments with varying USW parameters, ${\vec{\rho }}_{i}={\left(\omega_{i},\theta_{i},t_{i},\tau_{i},P_{i}\right)}^{T}$, and measurement results of the USW joint properties, ${\vec{R}}_{i}={\left(\sigma_{i},\varepsilon_{i},A_{i},\Delta h_{i}\right)}^{T}$, where $i=\bar{1,N}$ is the experiment number. The possible options for the mathematical description of the USW process, such as the relationship between such parameters and properties (structural and mechanical characteristics of the USW joints), were considered.

In the simplest case, the relationships between the mechanical properties of the USW joints can be represented as functional dependencies:$$\sigma_{UTS}=f_{\sigma }\left(\vec{\rho }\right),\;\varepsilon =f_{\varepsilon }\left(\vec{\rho }\right),\;A=f_{A}\left(\vec{\rho }\right),\;\Delta h=f_{\Delta h}\left(\vec{\rho }\right),$$
where $f\left(\vec{\rho }\right)$ are scalar functions of multiple arguments. In general, given that USW is a non-linear process, these functions should also be non-linear. In this case, the optimality condition (1) leads to a system of non-linear inequalities:(2)$$\{\begin{array}{c} \sigma_{\min}<f_{\sigma }\left(\vec{\rho }\right)<\sigma_{\max},\\ \varepsilon_{\min}<f_{\varepsilon }\left(\vec{\rho }\right)<\varepsilon_{\max},\\ A_{\min}<f_{A}\left(\vec{\rho }\right)<A_{\max},\\ \Delta h_{\min}<f_{\Delta h}\left(\vec{\rho }\right)<\Delta h_{\max}. \end{array}$$

If factors affecting the random outcome of the experiments and the measurement errors of the required USW joint properties are neglected (or their averaged values are used), the problem of determining the optimal USW parameters is reduced to interpolation of *N* measurements in such a multidimensional space, i.e., defining the $f\left(\vec{\rho }\right)$ functions that take values of the ${\vec{R}}_{i}$ required USW joint properties for the ${\vec{\rho }}_{i}$ USW parameters:$$f_{\sigma }\left({\vec{\rho }}_{i}\right)=\sigma_{i},\;f_{\varepsilon }\left({\vec{\rho }}_{i}\right)=\varepsilon_{i},\;f_{A}\left({\vec{\rho }}_{i}\right)=A_{i},\;f_{\Delta h}\left(\vec{\rho }\right)=\Delta h_{i},$$
and a search for such a range of Ω values of the $\vec{\rho }\in \Omega$ parameter vector within which the property functions satisfy the system of inequalities (2). The following methods can be applied to interpolate functions in a multidimensional parameter space: triangulation with linear interpolation, inverse distance to a power gridding, basis functions, polynomial and kriging functions, etc. [33,34,35].

If the ${\vec{\Delta }}_{R}={\left(\delta_{\sigma },\delta_{\varepsilon },\delta_{A},\delta_{h}\right)}^{T}$ accuracy of the measuring equipment and the ${\vec{\Delta }}_{\rho }={\left(\delta_{\omega },\delta_{\theta },\delta_{t},\delta_{\tau },\delta_{P}\right)}^{T}$ possible random deviations of the given USW parameters from the real values are known, the task of finding the optimal USW modes is reduced to approximating *N* measured values in a multidimensional space of the process parameters, i.e., defining such $f\left(\vec{\rho }\right)$ functions that take values in the intervals
$$\begin{array}{c} \sigma_{i}-\delta_{\sigma }\leq f_{\sigma }\left(\vec{\rho }\right)\leq \sigma_{i}+\delta_{\sigma },\\ \varepsilon_{i}-\delta_{\varepsilon }\leq f_{\varepsilon }\left(\vec{\rho }\right)\leq \varepsilon_{i}+\delta_{\varepsilon },\\ A_{i}-\delta_{A}\leq f_{A}\left(\vec{\rho }\right)\leq A_{i}+\delta_{A},\\ \Delta h_{i}-\delta_{h}\leq f_{\Delta h}\left(\vec{\rho }\right)\leq \Delta h_{i}+\delta_{h}, \end{array}$$
varying the USW parameters in the $\vec{\rho }\in \left({\vec{\rho }}_{i}\pm {\vec{\Delta }}_{\rho }\right)$ range, and searching for such a Ω range of values of the $\vec{\rho }\in \Omega$ parameter vector within which the property functions satisfy the system of inequalities (2). To solve the problem in this formulation, a regression analysis method based on least squares is typically applied [36,37]. For multi-extremum functions, piecewise approximation algorithms and algorithms based on component-wise spline approximation are recognized as the most efficient [16,17,38]. Representation of a function of multiple variables using linear operations and superposition of functions of one variable was enhanced in the theory of neural networks [39,40,41].

Taking into account that some properties of the same object are determined, which are characterized by their correlation, the mathematical description of the dependence of the mechanical properties on the USW parameters should be represented as a nonlinear system with multiple inputs and outputs:(3)$$\vec{R}=F\left(\vec{\rho }\right),$$
where $F\left(\vec{\rho }\right)$ is a vector function of multiple variables. The optimality condition can be written as a decision rule:(4)$$Q\left(\vec{\rho }\right)=\{\left(\sigma_{\min}<\sigma_{UTS}<\sigma_{\max}\right)\wedge \left(\varepsilon_{\min}<\varepsilon <\varepsilon_{\max}\right)\wedge \;\wedge \left(A_{\min}<A<A_{\max}\right)\wedge \left(\Delta h_{\min}<\Delta h<\Delta h_{\max}\right);{\left(\sigma_{UTS},\varepsilon ,A,\Delta h\right)}^{T}=F\left(\vec{\rho }\right)\},$$
where ∧ is the “AND” logical operation. To verify the decision rule (4), the following steps must be performed: (a) for a given set of parameters, determine the properties according to expression (3); and (b) check the property values according to the decision rule (4). If all inequalities are simultaneously true, the response of the rule is set to “true”; otherwise, it is set to “false”.

The problem of determining the optimal USW conditions with both known ${\vec{\Delta }}_{R}$ accuracy and ${\vec{\Delta }}_{\rho }$ deviations is reduced to approximating *N* vector measurements in a multidimensional space of the USW parameters, i.e., determining an $F\left(\vec{\rho }\right)$ function that takes a vector value in the ${\vec{R}}_{i}\pm {\vec{\Delta }}_{R}$ range when varying a parameter in the $\vec{\rho }\in \left({\vec{\rho }}_{i}\pm {\vec{\Delta }}_{\rho }\right)$ range and searching for an Ω range of values of the $\vec{\rho }\in \Omega$ parameter vector within which the decision rule (4) is true. Computer simulation in this formulation can be carried out by according to the above-mentioned methods of approximating empirical data by scalar functions, together with the design of the studied object architecture. If the internal structure of the object remains a “black box”, problems of this type can be solved using neural network simulation (NNS) methods [40,42,43].

## 3. Research Methodology

The standard research methodology for analysis of experimental data includes several main stages: (i) design of the experiment, (ii) testing, (iii) preprocessing of the experimental data, (iv) constructing a model or a set of object models, and (v) verification of the model [40,44,45].

Designing an experiment can significantly reduce the research complexity compared to a full-factor experiment, especially when the number of input parameters exceeds two. The existing approaches to solving this problem are summarized in [45]. In this study, we used the Taguchi method, which is described below in Section 4.

At the stage of preprocessing the experimental results, the following problems were solved:Analysis of the reliability of the results of each experiment and, if necessary, rejection of incorrect data;Analysis of the completeness of the summarized information or, with a small sample, determination of limitations on the simulation results; andAnalysis of the data dimensions.

The results were statistically processed according to four test coupons with the confidence interval calculation. Then, according to the accepted confidence interval, one value was discarded. Therefore, the values for three test coupons were used for analysis.

Another problem associated with almost all data processing and analysis methods is their varying dimensions. It is necessary to choose metrics among known or newly developed metric in both parameter and property spaces. The simplest solution to this problem is to convert both spaces to dimensionless spaces. Such transformations essentially constitute normalization of the original data of the form:$$\bar{z}=\frac{z-\min\left(z\right)}{\max\left(z\right)-\min\left(z\right)},\;z=\left(\max\left(z\right)-\min\left(z\right)\right)\bar{z}+\min\left(z\right),$$
where *z* and $\bar{z}$ are the normalizing and normalized values, respectively; and min(z) and max(z) are the minimum and maximum levels of the normalized value, respectively. A priori known ranges or ranges calculated from available data can serve as the minimum and maximum levels. For neural networks, the data normalization stage should provide the first layer of neurons. However, the training stage becomes more complicated in the case of a large difference in the input data values, and preliminary normalization is carried out to accelerate the process.

According to [46], it is sufficient to choose a neural network with one hidden layer and a small number of neurons to solve the regression problem. With three input and four output parameters, the minimum number of neurons in the hidden layer should be at least four. To ensure bounded values at the neural network output, it is necessary to select a bounded function as the activation function, which can be a sigmoid, hyperbolic tangent, etc. The activation function of the hidden layer should be preset to achieve a better approximation or a faster learning rate (irrelevant in the case of a small training sample).

It is generally accepted that a neural network needs to be trained on sufficiently large and representative sample that covers all classes and represents the variability within each class for classification problems. For approximation problems, the meaning of the training sample remains the same, but the sample is represented by single values for individual points in the parameter space. Therefore, the training sample should be formed for the neural network. Accordingly, an assumption is made about the distribution law for the ${\vec{\Delta }}_{R}$ property measurement errors and the ${\vec{\Delta }}_{\rho }$ parameter deviations. Then, a training sample is synthesized in both parameter and property spaces by simulation of a sequence of pseudo-random vectors in the [${\vec{R}}_{i}-{\vec{\Delta }}_{R}$, ${\vec{R}}_{i}+{\vec{\Delta }}_{R}$] and [${\vec{\rho }}_{i}-{\vec{\Delta }}_{\rho }$,${\vec{\rho }}_{i}+{\vec{\Delta }}_{\rho }$] regions, as well as by summing the initial data in this sequence.

In the final stage, the numerical models are verified. With a sufficiently large (redundant) training sample, a testing sample is selected and cross-checked [40,41,43]. By comparing the sample characteristics and the simulation results, a conclusion can be drawn about its errors. Either the cross-validation, jackknife, bootstrap, hold-out, or k-fold method is applied depending on the testing sample formation procedure [34,40,43,47]. For approximation problems, when there are single measurements in the input data space of the training sample, such verification methods cannot be implemented in most cases. To assess the adequacy of the models, additional full-scale experiments are required. For example, an experiment is conducted to test USW parameters that tend to be optimal according to the results of a number of models; however, their non-optimality is revealed according to other models.

## 4. Materials and Methods

The following components were used to manufacture USW lap joints: 50 × 20 × 2 mm PEEK plates made from “PEEK 770PF” powder (Zeepeek, Changchun, Jilin Province, China) by plunge casting, a PEEK-based prepreg in the form of a tape of unidirectional CFs (Toray Cetex TC1200, thickness of 140 µm)(Morgan Hill, CA, USA), and a PEEK film (Victrex, Aptiv 2000, thickness of 250 µm) (Lancashire, UK) as an ED.

The USW lap joints were formed using a “UZPS-7” ultrasonic welding machine (SpetsmashSonic LLC, Voronezh, Russia). The plates to be welded were placed in a fixing clamp, which prevented their mutual movement. The dimensions of the sonotrode that was used to apply US vibrations to the overlapped adherends were 20 × 20 mm. The test coupons included five layers, namely two outer PEEK layers, a central prepreg layer, and two intermediate EDs (Figure 1).

Tensile tests of the USW joints were performed according to ASTM D5868. The tests were carried out with an “Instron 5582” electromechanical tensile testing machine (Instron, Norwood, MA, USA). The cross-head speed was 1 mm/min. To minimize misalignment during the tests, gaskets made of identical-thickness PEEK plates were installed in wedge grips at the test coupon gripping. The USW joint thinning was measured with a micrometer.

The cross-sectional structure of the USW joints was analyzed using a “Neophot 2” optical microscope (OM; Carl Zeiss, Jena, Germany).

In the experiment design, levels of all factors (the USW parameters) and their possible changing ranges were preset:The USW duration (*t*) range was set between 800 and 1200 ms because it was not possible to join the PEEK plates at lower values; however, the prepreg could be damaged at higher levels, resulting in faulty USW joints;Ranges of the clamping pressure (*P*) and its (holding) duration after applying US vibrations (*τ*) were determined based on the technical characteristics of the USW machine, as well as by visual control of the USW joints.

The data on the mechanical characteristics were ranked according to the Taguchi method in the form of a table of combining all factors and their levels (Table 1). Each level and factor were presented three times. Thus, nine types of USW joints were fabricated for mechanical testing, the results of which reflected the following characteristics (Table 2): (1) $\sigma_{UTS}$ (MPa), (2) *ε* (mm), (3) *A* (N·m), and (4) Δ*h* (mm).

The Taguchi method [15] was implemented to identify the key properties of the USW joints. The means statistics were calculated, and the dependences of the physical and mechanical characteristics were plotted based on the USW parameters (Figure 2).

According to Figure 2, an increase in the clamping duration after US vibrations (*τ*) from 500 to 1500 ms was accompanied by a minimal change in ultimate tensile strength. In this case, the clamping pressure (*P*) and duration of US vibrations (*t*) were the most significant factors. The next analyzed parameter was elongation at break (Figure 2b). The clamping pressure (*P*) and duration of US vibrations (*t*) were the most influential factors, whereas the clamping duration after US vibrations (*τ*) exerted a much lower influence. Dependences of the work of strain (*A*) values from the USW parameters showed the same trend as elongation at break (Figure 2c). In terms of USW joint thinning, the clamping duration after US vibrations (*τ*) was characterized by a more negligible effect (Figure 2d). Duration of US vibrations (*t*) exerted the maximum influence.

Based on the obtained results, the USW parameters were ranked. In this stage, we attempted to obtain the maximum mechanical properties. The scoring system is summarized in Table 3. The factor that had the greatest impact was marked with the lowest number. The *t* parameter was found to have made the greatest contribution to the increase in mechanical properties (rational value among the studied values was 1200 ms). The *P* parameter had slightly less effect (its rational level was 3.0 atm). The *τ* parameter was characterized by a minimal impact; therefore, its maximum value of 1500 ms was taken as “applied”.

According to the results obtained using the Taguchi method, the maximum mechanical characteristics (both ultimate tensile strength and elongation at break) were provided by mode #9 (*P* = 3.0 atm, *τ* = 1200 ms). However, as shown below in the Discussion section, mode #9 gave rise to melting of the prepreg and partially damaged CFs. Therefore, an alternative method was required to determine the optimal combination of USW parameters more objectively, primarily by taking into account the structural characteristics.

## 5. Research Results

The minimum and maximum values were calculated (Table 4) based on the available training sample (Table 1 and Table 2), which were used for normalization. The normalized values of the USW parameters and the USW joint properties are presented in Table 4.

The training samples was formed on the basis of the assumption of the ${\vec{\rho }}_{i}$ and ${\vec{R}}_{i}$ distribution law of uniformity with ${\vec{\Delta }}_{R}$ accuracy and ${\vec{\Delta }}_{\rho }$ deviation of 5%. The training sample synthesis algorithm included synthesis of the sample using a pseudo-random number generator with a uniform distribution law in the [−0.05, 0.05] range and adding them to the normalized values of the USW parameters and the USW joint properties. A set of 54 vectors of US welding parameters were generated, with 54 vectors of corresponding properties. The training sample consisted of eight pairs of randomly selected vectors. The remaining 46 pairs composed the training sample. 

The NNS method was implemented using the MathLab^®^ software package, which includes tools for their synthesis, training, and analysis. All models relied on direct propagation with one hidden layer (example shown in Figure 3). The number of neurons in the hidden layer and the applied activation functions was varied. The use of the linear activation function resulted in designing untrained networks. Therefore, only networks with a hyperbolic tangent as an activation function of both hidden and output layers are discussed below.

The following methods were tested as training methods: (i) Levenberg–Marquard, (ii) gradient descent, (iii) random increments, (iv) successive increments, (v) Fletcher–Powell coupled gradients, and (vi) Bayesian regularization [41,43,47]. Bayesian regularization achieved the best results and the highest convergence rate. Therefore, it was implemented for further analysis. According to the results of training in the normalized space, the minimum mean square error (*MSE*) and the *R* determination coefficient were estimated, and correlation schemes of the simulated and real data were drawn. The networks with the best learning outcomes are presented in Table 5 and Figure 4.

Then, the synthesized artificial neural networks (ANNs) were used to calculate the USW joint properties in the three-dimensional space of the USW parameters. The simulation range was chosen according to the previously accepted normalization limits (Table 5). Reliability analysis of the models was carried out according to the sections of the three-dimensional space of the USW parameters (Figure 5). The properties vector $\vec{R}$ was calculated according to expression (3), and $F\left(\vec{\rho }\right)$ was determined using ANNs 1–7. The values of the USW vector parameters were determined over the space sections by crossing two points with the specified properties. The USW duration was varied within the normalized range, whereas the clamping duration possessed the fixed value of *τ* = 1000 ms. The clamping pressure was calculated for each USW duration through the linear dependence of the following type: *P*(*t*) = (*P_i_* − (*t* − *t_i_*)/(*t_j_* − *t_i_*)), where *P_i_*, *t*_i_, and *t*_j_ are the values of parameters of the *i*-th and *j*-th experiments, respectively.

Based on the obtained results, it was found that after approximation, models 6 and 7 showed a number of excessive extrema, as shown in Figure 5e–f. In terms of the NNS, this phenomenon indicated their “overtraining”, including as a result of an excessive number of neurons.

On the other hand, Figure 5c,e show that for both low *t* and high *P* values, as well as for both high *t* and low *P* levels, the behavior trends of models 1–3 were opposite to those for models 4–7. This did not reflect that these results contradicted the physical meaning but that they required experimental verification. Note the excessive extremes for model 1 in Figure 5b,f. This result could be interpreted as an “undertrained” ANN. Therefore, only models 2–5 were further investigated.

Considering the results of the analysis of the structure and the fracture patterns of the USW joints, as well as based on general considerations about the relationship between the structure and mechanical properties, the choice of the threshold values was substantiated below (Table 6). For a better understanding of the used motivation, Figure 6 shows the stress–strain diagrams for the studied USW joints.

The following conclusions were drawn based on these data:The USW modes that provided low ultimate tensile strength levels of the USW joints were considered unacceptable. In particular, the lowest threshold level of 60 MPa was assigned by the authors. On the other hand, failure of the USW joints showed ultimate tensile strength values slightly higher than that for neat PEEK (~93 MPa), occurred over the base material but not in the fusion zone due to its over-strengthening. The most likely reason for of this phenomenon was damage and partial melting of the prepreg. Therefore, the maximum threshold of 93 MPa was assigned.Based on the same considerations, the modes providing brittle fracture at low elongation at break levels (*ε* < 2 mm) were excluded. On the other hand, fusion-zone over-strengthening caused strains with the formation of a “neck” in the base material at *ε* > 7 mm.Because the work of strain value combines both strength and ductility properties, the range of its acceptable values was assigned 150 N·m < *A* < 560 N·m by analogy with the above considerations.USW joint thinning was related to the pattern of the formed macro- and microstructure. In the case of USW joint-thinning values above 0.5 mm, the prepreg was melted, causing fracture of the CFs. On the other hand, the USW joint components, primarily the EDs, clearly did not melt when the USW joint-thinning level was less than 0.3 mm. In such cases, the USW joints were characterized by poor mechanical properties. Thus, the USW joint-thinning range was assigned as 0.3 mm < Δ*h* < 0.5 mm.

Optimality range analysis for the USW parameters was carried out according to the approved decision rule (4) over the entire volume under consideration. ANN simulation results were combined in a summary (collecting) isoline graph over the 3D space section. The optimal values of technological parameters were highlighted as blue-filled regions. Figure 7 shows examples of such graphs in the isolines for the space sections of the ANN’s parameters at the clamping duration after US vibrations of 1500 ms. Depending on the selected model, the following combinations of clamping pressure and duration of US vibrations could be distinguished: 

*Model 2.* The extended optimality region was oriented horizontally at *P* = 3.0 atm and *t* from 600 to 1000 ms, which was a fairly wide range of the values. The combination of P ~ 3.0 atm/t ~ 800 ms corresponding to USW mode #2 and resulting in improved mechanical properties should be considered the middle of this region.

*Model 3.* There were two optimality areas. The center of the first optimality area corresponded to *P* ~ 3.5 atm/*t* ~ 1000 ms, whereas the second one was observed at *P* ~ 1.5 atm/*t* ~ 750 ms. The USW parameters of the first region were close to mode #4 (*P* = 3.0 atm/*t* = 1000 ms), when sufficiently high mechanical properties of the USW joint were provided. The parameters of the second region were not experimentally verified in this research. Then, the prediction correctness of this ANN model was verified.

*Model 4.* In this case, there was only one optimality region, with the center at *P* ~ 3.0 atm/*t* ~ 1000 ms, which approximately reflected the prediction of the previous model (2) (experimentally studied USW mode #4).

*Model 5.* Similar to model 3, this model revealed two optimality areas. Their centers also roughly corresponded to those for model 3: (1) *P* ~ 3.2 atm/*t* ~ 1070 ms (close to mode #4); (2) *P* ~ 1.4 atm/*t* ~ 800 ms (not studied experimentally).

The following combinations were determined as optimal: (i) *P* = 3.0 atm/*t* = 800 ms (mode #2), (ii) *P* = 3.0 atm/*t* = 1000 ms (mode #4), and (iii) *P* = 1.5 atm/*t* = 800 ms (hereafter referred to as mode #10). According to the first two modes (not considering the varying *τ* values), the USW joints were fabricated and tested. According to the third combination (mode #10), new USW joints were manufactured. The results of their tests are analyzed in the Discussion section.

## 6. Discussion

Because the *τ* parameter showed the least effect according to the applied Taguchi method, it was not analyzed in detail when discussing the results. Consequently, Figure 7 shows data for the single *τ* value of 1500 ms.

According to the stress–strain diagrams (Figure 6), the highest strain–strength properties (within the assigned threshold values) were provided by USW modes #2 and #4. Micrographs of the cross section (fusion zone) of the corresponding USW joints are shown in Figure 8b,c. For mode #2, the prepreg remained undamaged (without any discontinuities). In the case of mode #4, the EDs were melted to a greater extent, but the prepreg also predominantly retained its integrity. These results were consistent with the above prediction.

Figure 9 presents photographs of several fractured lap joints. Note that regardless of the use of the compensating gaskets aimed at providing alignment of the test coupon under tension, a macroscopic bending developed due to enhanced stiffness of the central part of the lap. As a result, failure developed both due to shear stresses at the adherends’ interface (interlayer boundary) and bending induced in the main crack (nucleated at the weld edges). The details of this phenomenon are described in [48]. When the ED was undermelted (mode #1), the lap joint fractured close to the US weld edge (Figure 9a). The test coupon failed over the base polymer, with neck formation in the case of interlayer interface over-strengthening (mode #9, Figure 9c). When the optimal modes were employed (Figure 9b,d) the main cracks initially developed along the interlayer boundary, with subsequent rapture in the middle of the weld (due to lap-joint macroscopic bending). 

The reasons for the exclusion of the USW joints fabricated at *t* = 1200 ms (modes #7–#9) became apparent from the analysis of the micrographs of their fusion zone. Figure 8e–k indicates that such a high duration of US vibrations caused melting of the prepreg (which was also evidenced by USW joint thinning of more than 500 µm according to Table 2), damaging of CFs, as well as the formation of pores (due to the overheating). Such modes cannot be considered acceptable.

We also considered it necessary to provide the micrographs of the cross section of the test coupon fabricated according to mode #1 (*P* = 2.0 atm/*t* = 800 ms; Figure 8a,h), because this mode was in the optimality range. The presented images show the results of the mechanical tests, with the USW joint-thinning value indicating that ED melting occurred (Δ*h* = 0.42 mm). Therefore, the test coupon failed, with a slight elongation at break value of 3.4 mm an ultimate tensile strength of 76.2 MPa. These levels were within the accepted thresholds of the optimality condition. However, the fusion zone structure was heterogeneous. Figure 8h evidences pores and discontinuities. A prospective way to solve this problem is to reduce the clamping pressure (*P*). To this end, the same experiments were carried out using mode #10.

A USW joint formed according to the mode #10 was characterized as follow: $\sigma_{UTS}$ = 80.5 MPa; *ε* = 4.2 mm; *A* = 273 N·m; Δ*h* = 0.30 mm, satisfying the optimality condition. Analysis of the fusion zone micrograph (Figure 8d) showed that the prepreg retained its integrity. ED melting occurred even with short durations of US vibrations (*t*) of 800 ms due to the low clamping pressure (*P*) of 1.5 atm. Furthermore, the clamping pressure (*P*) minimized possible damage to the prepreg and CFs. Such a mode was predicted by both neural network models 3 and 5. Additionally, this mode was outside of the initial range in the design of the experiment.

The following result is of particular importance. Even if both mechanical and structural (USW joint-thinning (Δ*h*)) characteristics fell within the specified ranges, all structural changes were not taken into account, i.e., the degree and pattern of both ED and prepreg melting, possible damage of CFs, the formation and degree of porosity, etc. In particular, the optimality condition for the USW parameters could be accompanied by a local violation of the prepreg integrity (mode #4). However, structural characterization was a longer process than the assessment of the USW joint properties, requiring a much greater amount of the experimental data. Therefore, it was necessary to explicitly add parameters characterizing the integrity of the structural elements (the prepreg) to determine the optimal combination of USW parameters. They should be easily and quickly measured without a significant spread over different sections of the USW joints. We will conduct further research in this direction in the future.

As mentioned above, selection of optimal USW parameters is a key problem with respect to ensuring a lack of flaws, as well as high strength properties of permanent joints [49]. Of particular relevance are studies on design and variation of ED films, as they govern the melting and structural formation processes, as well as the strength properties of the laps [50,51,52]. The temperature field evolution of this highly non-stationary process requires attentive study, and computer simulation represents an efficient prediction tool [53,54]. In addition, the finite element method is a popular tool for numerical investigations of deformation processes occurring under mechanical loading of US-welded laminates [55]. Thus, the variety of process and parameters to be taken into account in the design of USW technology has expectedly attracted a machine learning approach, primarily ANNs [56,57,58,59,60].

The relevant papers on the application of ANNs to solve USW problems [56,58,61,62,63] focus on their synthesis with low approximation error (*MSE*, *RMSE*, etc.) of the known data, as well as prognosis of a single property of the joint. It should be stressed that many statistical methods are available to model a single parameter, offering competitive results with those of machine learning methods in terms of accuracy and robustness. We demonstrated that when modeling complex systems with a variety of output parameters, provided they are statistically correlated, an NNS can be adequately employed. The cited papers lack information on the optimality criteria of the USW parameters. 

In the current paper, the condition was formalized on achieving the optimality of the USW parameters by attaining the specified mechanical and structural properties in the form of an inequality system (1). The problem of selecting optimal USW parameters based on a small sample was solved in two steps: through continuous approximation of the material property vector with the use of (i) NSS and (ii) determination of the optimal parameters’ range limits. We demonstrated that special attention should be paid to the correctness of the simulation within the approximation and extrapolation ranges of experimental data rather than minimizing the approximation errors of known data (as investigated in numerous papers). 

As expected, computer simulation with a small training sample was limited due to the use of the simplest ANNs based on a single-layer perceptron, resulting in the need to synthesize several models [64,65]. Therefore, we needed assessment the adequacy of the obtained models, which was a somewhat subjective task. In this study, the solution to this problem was found at the level of assessing the correspondence to known or experimentally obtained physical patterns of the calculated values in the intervals between the measured data and in the extrapolation region. Prospects for further development of USW process simulation include the use of fuzzy logic methods to formalize the desired patterns at the ANN output and hybrid ANNs [66,67,68]. Another option is to form new ANNs that simulate the key structural characteristics, including by combining them into perceptron complexes with neural networks of the mechanical properties.

## 7. Conclusions

The following results were obtained in the present study:The definition of “the optimality combination of the USW parameters” was introduced as a condition for satisfaction of the inequality system (1). The mathematical problem of determining the optimal combination of USW parameters was formulated for the formation of USW lap joint with improved mechanical and structural properties.A methodology for studying the mechanical properties of USW lap joints was proposed based on the design of an experiment via the Taguchi method and experimental data approximation with the use of neural network simulation (NNS).Experiments were performed, and the threshold values of the optimality conditions for the USW parameters were chosen. NNS was accordingly carried out to determine their ranges.We demonstrated that according to the Taguchi method, the rational USW parameters corresponded to the *P* = 3.0 atm/*t* = 1200 ms combination. This evidence was based on consideration of the mechanical characteristics of the USW joints. However, intense prepreg melting and fracture of CFs occurred during the USW process. Therefore, this mode could not be considered optimal.We proved that verification of the computer simulation results obtained with a small sample can be carried out only by experimental analysis of their reliability. To obtain formal criteria to assess such facts, it is necessary to conduct additional studies.We analyzed the optimal combination of USW parameters proposed on the basis of NNS. Based on the example of two modes, we demonstrated that the developed optimality condition was insufficient and required correction, taking into account other significant structural characteristics of the formed USW joints. However, the NNS enabled specification of an extra area of USW parameters, which were not previously considered as optimal when designing the experiment. The NNS-predicted optimal US welding mode (*P* = 1.5 atm, *t* = 800 ms and *τ* = 1500 ms) ensured formation of lap joints with the required mechanical and structural properties (σ_UTS_ = 80.5 MPa; *ε* = 4.2 mm; *A* = 273 N·m; Δ*h* = 0.30 mm).The results of this study are related to the use of additional valuable structural characteristics, as well as attracting more complex neural network models. In terms the PEEK-CF layered composite fabrication with the use of US-welding, CF fabric prepregs are suitable for the design of “CF prepreg—PEEK composite” laminates.

## Figures and Tables

**Figure 1 materials-15-06939-f001:**
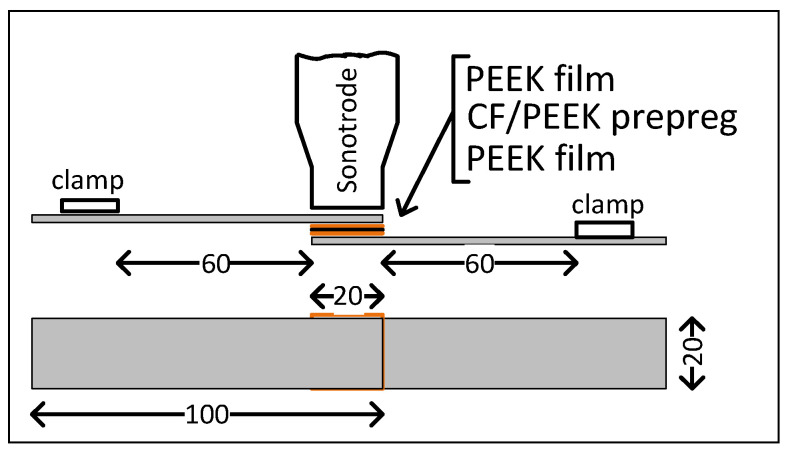
Schematic representation of the material stack for US welding.

**Figure 2 materials-15-06939-f002:**
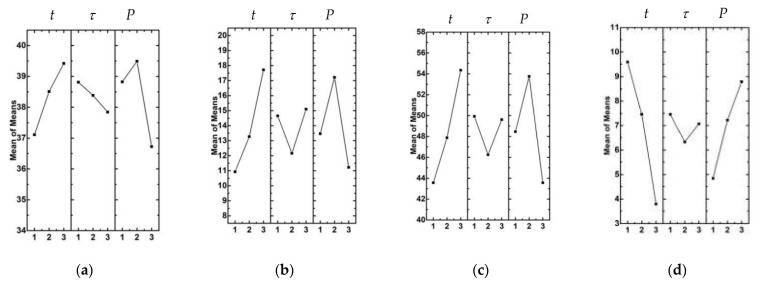
Dependencies of the mechanical and structural characteristics based on USW parameters: ultimate tensile strength (**a**), elongation at break (**b**), work of strain (**c**), and USW joint thinning (**d**).

**Figure 3 materials-15-06939-f003:**
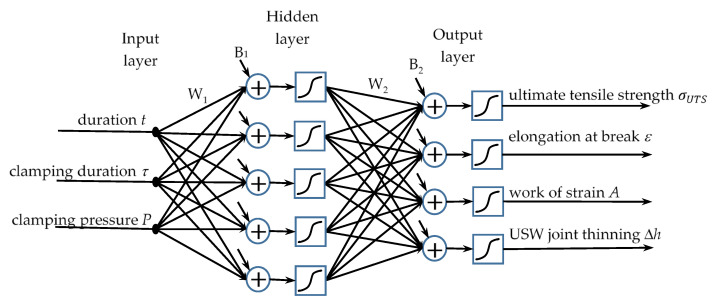
Architecture of artificial neural network (model 2). W—matrices of synaptic weights, B—vectors of bias weights.

**Figure 4 materials-15-06939-f004:**
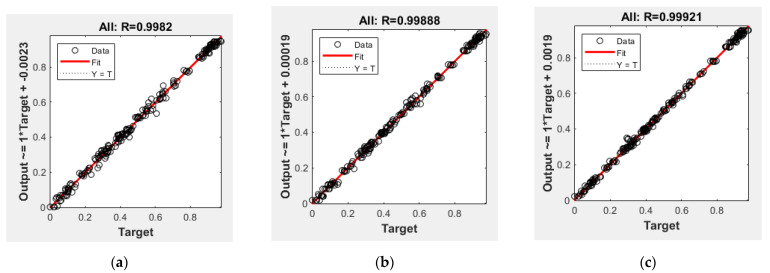
Correlation between the training sample data and the neural network output; neural network No.2 (**a**); No.4 (**b**); No.7 (**c**).

**Figure 5 materials-15-06939-f005:**
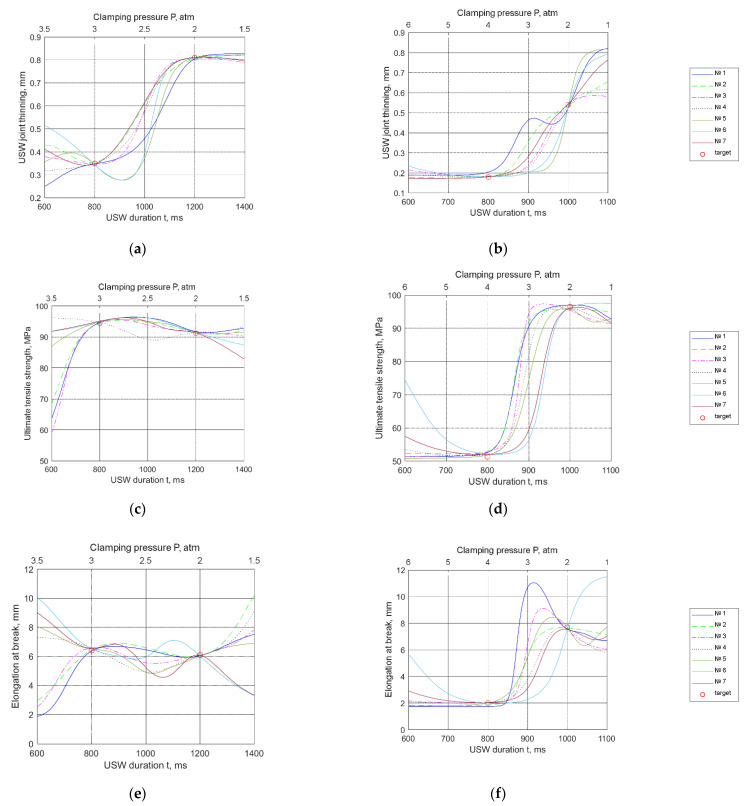
Simulation results over the space cross section of the USW parameters: (**a**,**c**,**e**) *τ* = 1000 ms; *P* = (3 − (*t* − 800)/400) atm; (**b**,**d**,**f**) *τ* = 1500 ms; *P* = (4 − 2(*t* − 800)/200) atm.

**Figure 6 materials-15-06939-f006:**
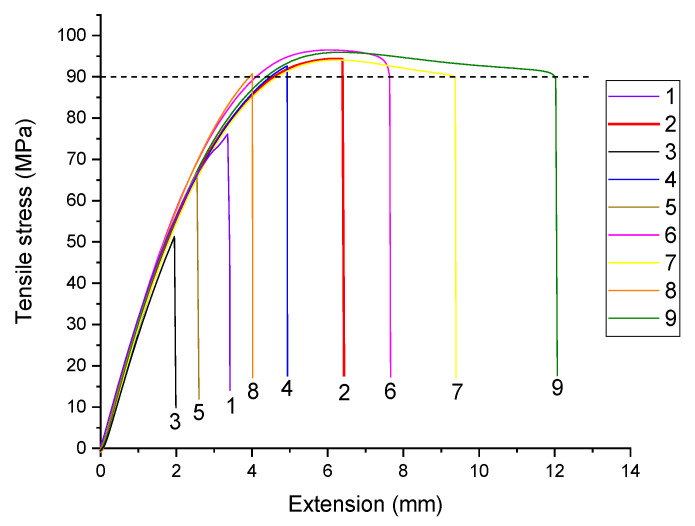
Stress–strain diagrams for the studied USW joints.

**Figure 7 materials-15-06939-f007:**
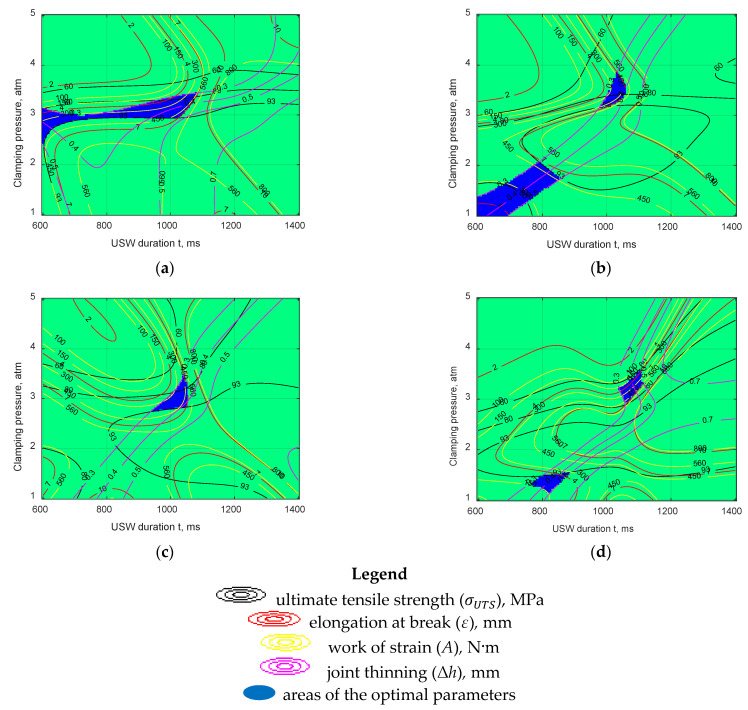
The graphs in the isolines for the sections of the space of ANN parameters; the clamping duration after US vibrations of 1500 ms: (**a**) model 2; (**b**) model 3; (**c**) model 4; (**d**) model 5.

**Figure 8 materials-15-06939-f008:**
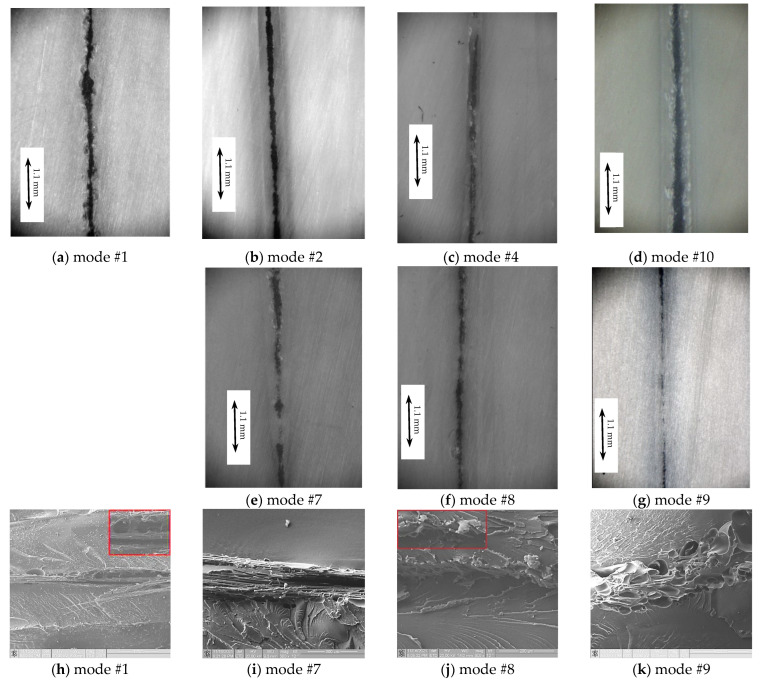
Macrostructure of the test coupons’ cross sections at the central part of the USW joints formed using the modes according to Table 1 (**a**–**g**) and micrographs of their fusion zones (**h**–**k**).

**Figure 9 materials-15-06939-f009:**
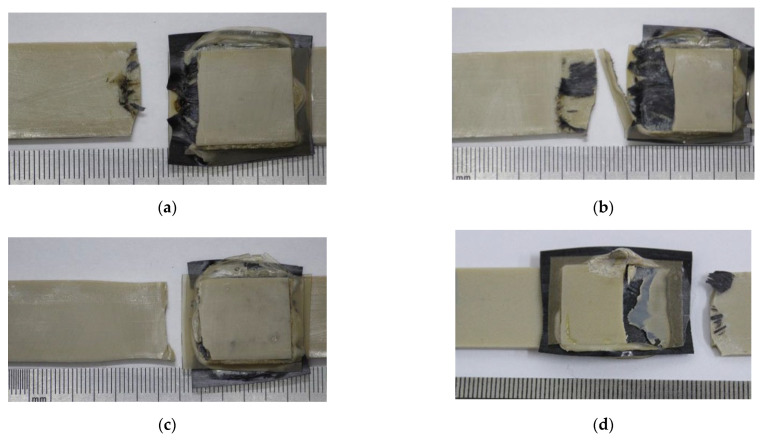
Fracture patterns of lap joints. Mode #1 (**a**); mode #4 (**b**); mode #9 (**c**); mode #10 (**d**).

**Table 1 materials-15-06939-t001:** Combination of the USW parameters and their levels (according to the Taguchi table in L9 format for a three-factor experiment).

Experiment Number	Level/Factor		
	USW Duration (*t*), ms	Clamping Duration after US Vibrations (*τ*), ms	Clamping Pressure (*P*), atm
1	1/800	1/500	1/ 2
2	1/800	2/1000	2/3
3	1/800	3/ 1500	3/4
4	2/1000	1/500	2/3
5	2/1000	2/1000	3/4
6	2/1000	3/1500	1/2
7	3/1200	1/500	3/4
8	3/1200	2/1000	1/2
9	3/1200	3/1500	2/3

**Table 2 materials-15-06939-t002:** Mechanical and structural characteristics of the USW joints for various USW parameters.

Experiment (Mode) Number	Ultimate Tensile Strength $\left(\sigma_{UTS}\right)$, MPa	Elongation at Break (*ε*), mm	Work of Fracture (*A*), N·m	USW Joint Thinning (Δ*h*), mm
1	76.2 ± 0.4	3.4 ± 0.1	156.2 ± 12.2	0.42 ± 0.02
2	94.4 ± 0.8	6.4 ± 0.2	424.7 ± 24.5	0.35 ± 0.02
3	51.3 ± 0.5	2.0 ± 0.1	51.7 ± 4.6	0.18 ± 0.01
4	92.6 ± 0.5	4.9 ± 0.2	284.3 ± 15.3	0.38 ± 0.02
5	66.9 ± 0.4	2.6 ± 0.2	95.6 ± 6.7	0.32 ± 0.02
6	96.5 ± 0.5	7.7 ± 0.3	558.6 ± 37.7	0.54 ± 0.02
7	94.1 ± 0.6	9.4 ± 0.3	694.4 ± 38.1	0.48 ± 0.03
8	91.2 ± 0.8	6.1 ± 0.4	386.3 ± 24.4	0.81 ± 0.03
9	95.9 ± 0.7	12.1 ± 1.0	958.0 ± 52.1	0.60 ± 0.02

**Table 3 materials-15-06939-t003:** Influence of the USW parameters on the joint characteristics.

Property	USW Duration (*t*), ms	Clamping Duration after US Vibrations (*τ*), ms	Clamping Pressure (*P*), atm
Ultimate tensile strength	2	3	1
Elongation at break	1	3	2
Work of strain	1	3	2
USW joint thinning	1	3	2
Total	5	12	7
“Best” level	1200	1500	3

**Table 4 materials-15-06939-t004:** Parameter and property values based on analysis results.

Characteristic, z	Experimental Data Range		Normalization Range	
	min (z)	max (z)	Min	Max
*t*, ms	800	1200	600	1400
*τ*, ms	500	1500	400	2500
*P*, atm	2	4	1	5
$\sigma_{UTS}$, MPa	51.3	96.5	20	100
*ε*, mm	2.0	12.1	1	14
*A*, N·m	51.7	950	30	1230
Δ*h*, mm	0.26	0.87	0.1	0.9

**Table 5 materials-15-06939-t005:** Neural network parameters and the learning outcomes.

Model (Neural Network) Number	Number of Neurons	*R*	*MSE*
1	4	0.99773	1.1678 × 10^−4^
2	5	0.9982	6.2615 × 10^−5^
3	6	0.99857	4.9248 × 10^−5^
4	7	0.99888	3.4767 × 10^−5^
5	9	0.99911	7.0265 × 10^−5^
6	11	0.99918	4.3210 × 10^−5^
7	13	0.99921	5.0921 × 10^−5^

**Table 6 materials-15-06939-t006:** Threshold values for the optimal conditions.

Characteristic	Thresholds	
	Min	Max
$\sigma_{UTS}$, MPa	60	93
*ε*, mm	2.0	7.0
*A*, N·m	150	560
Δ*h*, mm	0.3	0.5

## Data Availability

The data presented in this study are available on request from the corresponding author. The data are not publicly available due to confidential disclosure reasons.

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
