# Peer review of "Optimizing Ultrasonic Welding Parameters for Multilayer Lap Joints of PEEK and Carbon Fibers by Neural Network Simulation"

_materials, 2022, doi:10.3390/ma15196939_

Round 1
Reviewer 1 Report
Abstract
Can you be more specific in the abstract (report quantitative outcomes)?
Introduction
1. At the end of the introduction, can you explain what is the purpose of the multi-layered welded interface? What goal do you aim to achieve, as opposed to using “traditional” energy directors (flat film or protrusions)? What are you aiming to improve based on the literature?
2. It would be useful to add a schematic of the material stack. From the description, it appears you are joining two unreinforced PEEK samples in a lap configuration, with 3 layers (two PEEK films and one CF/PEEK prepreg layer), but a visual representation would be helpful.
Problem statement
1. I understand the frequency being fixed, but keeping the amplitude constant is unusual (given it can be varied with US welders easily). Can you justify why it was kept constant?
2. Given the focus placed on the multilayered interface, it would have been interesting to see input parameters related to the layers at the interface.
3. It is unclear why some of the key strain-strength characteristics were chosen. Can you provide explanations or references supporting your selection?
4. pp. 3-6: Can you provide more details about the physical experiments (number of experiments, specific welding parameters for each experiment)? The need for a sufficiently large sample size is mentioned, but it’s never clear how many experiments will be carried out.
Materials, methods and research results
1. I would recommend to have separate sections for “Materials and methods”, then “Research results” (since section 5 is already named “Research results”).
2. Does the thickness of the layers at the joint matter? As mentioned for the introduction, it would be useful to add a schematic of your joints.
3. What is the frequency and amplitude of the welder?
4. Table 1 and Table 2: did you repeat each experiment more than once? How did you estimate uncertainty on the values presented in Table 2?
5. Table 2: in the text, you mentioned tensile shear strength, not ultimate tensile strength. How were the strength values calculated in Table 2? Can you explain why your numbers are much higher than welded CF/PEEK single lap joints found in the literature? What did the fracture surfaces look like?
6. p. 8: the effect of the welding parameters on each output is pretty clear from the Taguchi experiment. This is not quite clear why machine learning is needed. Please, elaborate.
Research results
1. What was your actual training sample size and testing sample size?
2. Figure 4: can you comment on the possibility of overfitting?
3. Figure 5: describe in more detail how those curves were obtained?
4. p. 12, bullet point 1: are you saying that failure occurred in the adherends and not at the joint in some cases? This needs to be discussed in more details. It means that in some cases, you used tensile strength (of the adherend) and in other cases, shear strength at the joint as one of the outputs. Can this affect your optimization model?
5. Figure 7: Can you should experimental results to confirm those areas of optimal parameters (if they were not originally part of your Taguchi experiment)?
Conclusions
1. Last bullet point mentions that “Its verification by a full-scale experiment at P = 1.5 atm, t = 800 ms and tau = 1,500 ms confirmed the correctness of the analysis of the adopted neural network models.”
I don’t clearly see where this data/outcome was presented in the paper. Can you out more emphasis in the text?
Author Response
Esteemed reviewer, we would like to express warmest gratitude for the attentive reading and thorough analysis of the manuscript. All your comments and remarks are relevant and valuable. While answering your questions we have found a lot of important information. The revision of the manuscript was extremely fruitful both for making the text more clear and to stress important results and contrast with literature data. Moreover, we have established a couple of advanced problems to be solved in our forthcoming studies.
Please find below the answers for all your comments. All the required changes have been made or added to the text. Let us thank you once again for helping us to improve the manuscript.
Abstract
Can you be more specific in the abstract (report quantitative outcomes)?
The required information has been added to the Abstract.
Introduction
- At the end of the introduction, can you explain what is the purpose of the multi-layered welded interface? What goal do you aim to achieve, as opposed to using “traditional” energy directors (flat film or protrusions)? What are you aiming to improve based on the literature?
Esteemed reviewer! The required information has been added to the Introduction section.
- It would be useful to add a schematic of the material stack. From the description, it appears you are joining two unreinforced PEEK samples in a lap configuration, with 3 layers (two PEEK films and one CF/PEEK prepreg layer), but a visual representation would be helpful.
Esteemed reviewer, thank you for the relevant piece of advice. The required information has been added.
Problem statement
- I understand the frequency being fixed, but keeping the amplitude constant is unusual (given it can be varied with US welders easily). Can you justify why it was kept constant?
Esteemed reviewer! The available US welding machine was designed without variation of amplitude. For sure, it is of interest to increase the amplitude; however, the frequency is to be decreased. We do hope to study this issue in the future. For this reason the sonotrod vibration amplitude might be used as an input parameter when constructing a neural network model.
- Given the focus placed on the multilayered interface, it would have been interesting to see input parameters related to the layers at the interface.
Esteemed reviewer! You have raised very interesting issue, that was out of parametric study in this paper. We did change the thickness of EDs, but it was not a matter of neural network research. We do plan to design prepreg structure with variation of CF-fabric volume fraction. The composition and porosity of ED is a parameter to be varied as well. Your idea has been taken into account and the number of input parameters has been increased.
- It is unclear why some of the key strain-strength characteristics were chosen. Can you provide explanations or references supporting your selection?
Esteemed reviewer, this remark is of particular relevance. For sure, the value of Lap Shear Strength (LSS) is mostly employed for numerical characterization of US-welded lap-joints. Since we welded the neat PEEK workpieces, the failure did not happen along the interface all the time. In order to make the result comparable, the ultimate strength was the parameter under analysis. AS soon as the long fiber reinforced adherends will be investigated, the LSS will be the key estimated parameter. In addition, the failure load was employed to characterize the lap joint strength in [Li, Y., Lee, T. H., Wang, C., Wang, K., Tan, C., Banu, M., & Hu, S. J. (2018). An artificial neural network model for pre-dicting joint performance in ultrasonic welding of composites. Procedia CIRP, 76, 85–88. doi:10.1016/j.procir.2018.01.010].
- 3-6: Can you provide more details about the physical experiments (number of experiments, specific welding parameters for each experiment)? The need for a sufficiently large sample size is mentioned, but it’s never clear how many experiments will be carried out.
Esteemed reviewer, we are sorry for using the same term for specifying the test coupons and data used in ANN-modeling. We have renamed the former by a “test coupon” exactly. In addition, the data on number of used test coupons is given in Page 5 “Statistical processing of the results was carried out according to four samples with the confidence interval calculation. Then, according to the accepted confidence interval, one value was discarded. So, the values for three test coupons were used for analysis”.
Materials, methods and research results
- I would recommend to have separate sections for “Materials and methods”, then “Research results” (since section 5 is already named “Research results.
Esteemed reviewer! It was our fault to leave the term “research results” in the name of the Section 5. The due correction has been made.
- Does the thickness of the layers at the joint matter? As mentioned for the introduction, it would be useful to add a schematic of your joints.
Esteemed reviewer! The schematic has been added, many thanks for the brilliant idea. The thickness of the ED was varied form 250 up to 750 mm; however, the best results were obtained at the lowest thickness. However, it was not a part of parametric studies. The thickness of the prepreg was fixed, since it was a piece of industrial product. In the current studies, the prepreg is fabricated from CF-fabric. It will be the matter of our forthcoming investigation.
- What is the frequency and amplitude of the welder?
The frequency of the US-welder was constant – 22 kHz, that is similar to most of the cited references. The vibration amplitude was constant as well and equal to 30 mm . Unfortunately, the in the supplied design were can not vary the amplitude.
- Table 1 and Table 2: did you repeat each experiment more than once? How did you estimate uncertainty on the values presented in Table 2?
Esteemed reviewer! We are sorry for not specifying the experimental data scatter. We just presented the average numbers while we must give the scatter information. As far as was noticed above, at least 3 specimens (test coupon) of each type was fabricated and tested (we repeated each experiment 3 times). The required information has been added to the Table 2. Thank you for this valuable remark.
- Table 2: in the text, you mentioned tensile shear strength, not ultimate tensile strength. How were the strength values calculated in Table 2? Can you explain why your numbers are much higher than welded CF/PEEK single lap joints found in the literature? What did the fracture surfaces look like?
Esteemed reviewer! The data in Table 2 specifies ultimate tensile strength. The reason was briefly explained above. When calculating the LSS, the applied load was divided by adherends’ overlapping area. When calculating the ultimate strength, the load was divided by cross-section area (that is smaller). For this reason the UTS value was higher in contrast with traditionally analyzed LSS values for US-welding of PEEK-composites. Let me stress once again that when US-welding neat PEEK the interface is enforced by CFs. This prevents traditional interlayer pattern of failure.
The appearance of some fractured test coupons have been added to the manuscript. They evidently illustrate the fact that failure could not follow the shear mechanism.
- p. 8: the effect of the welding parameters on each output is pretty clear from the Taguchi experiment. This is not quite clear why machine learning is needed. Please, elaborate.
Esteemed reviewer, the Taguchi method is not an approach for finding out optimum parameters, it is an experiment planning method. Its key idea is to illustrate and rank the technological parameters by the level of their significance. However, it does not characterize the intermediate values as well as one located out of the analyzed range. The key idea of this paper is to develop and test a method for the forecasting US-welding parameters not specified in the design of the experiment. The necessity of applying the machine learning was explained in the problem statement section of the paper. Based on your remark we have changed the term “optimum” onto the “best”. Thank you for this relevant comment.
Research results
- What was your actual training sample size and testing sample size? Esteemed reviewer! We are sorry for incorrect translation of the text. We have made the due corrections and add them to the revised manuscript.
- Figure 4: can you comment on the possibility of overfitting?
Esteemed reviewer! Thank you for the relevant remark.
The presented results are evident for the small sample (9 elements) expanded with additionally generated sample (at the variance level of 5%). The attained values of MSE и R are related exactly to the small size of the sample. The 5% level of variance was selected being based on the experimentally measured values of the strength-stress properties. In this statement, the approximation problem is to be solved by selecting the appropriate architecture of a Neural Network. It is then controlled within the entire approximation space as well as in the forecasting zone. The overfitting is not evident at the data presented in figure 4. However, curves in figure 5 illustrate the exact manifestation of the overfitting effect.
- Figure 5: describe in more detail how those curves were obtained?
The following explanation has been added to the revised manuscript (page10). "The properties vector was calculated according to the expression (3), where the was determined with the use of ANN 1-7. The values of the USW vector parameters were determined over the space sections by crossing two points with the specified properties. The USW duration was varied within the normalized range, while the clamping duration had the fixed value t = 1,000 ms. The clamping presure was calculated for each USW duration through the leaner dependence of the following type P(t) = (Pi – (t – ti)/(tj - ti)), where Pi, ti, tj – the values of parameters of the i-th and j-th experiments, respectively".
Note that the formula (4) was slightly corrected.
- p. 12, bullet point 1: are you saying that failure occurred in the adherends and not at the joint in some cases? This needs to be discussed in more details. It means that in some cases, you used tensile strength (of the adherend) and in other cases, shear strength at the joint as one of the outputs. Can this affect your optimization model?
Esteemed reviewer! The comment is exactly relevant. In fact, we use only the ultimate strength to characterize the failure stress, regardless the fracture pattern. We suggest that this allowed us the optimization model to be affected. The lap-joint failure out of the joint meant over-strengthening of the lap interface due to carbon fiber (prepreg) damaging. This phenomena is to be excluded. Another limiting case was too low UTS value. It was usually related to undermelting of EDs. However, the undermelting of the EDs was accompanied by low Dh value. For this reason for parameters were employed for specifying the optimality ranges.
- Figure 7: Can you experimental results to confirm those areas of optimal parameters (if they were not originally part of your Taguchi experiment)?
Esteemed reviewer! Thank you for the relevant comment. Since some of the ANN-predicted values of the USW-parameters were close to ones used at design of experiment by the Taguchi method (the modes #2 and 4), we did not fabricated the lap-joint according to them. However, the mode #10 was even out of the range used at the Taguchi-based planning. This is an exact manifestation of predicting ability of the proposed ANN-method.
Conclusions
- Last bullet point mentions that “Its verification by a full-scale experiment at P = 1.5 atm, t = 800 ms and tau = 1,500 ms confirmed the correctness of the analysis of the adopted neural network models.” . I don’t clearly see where this data/outcome was presented in the paper. Can you out more emphasis in the text?
Esteemed reviewer! Thank you for the comment and the proposed idea. We have additionally emphasized the importance of this result in the text of the revised manuscript (in the Discussion section).
Reviewer 2 Report
- It is necessary to make a connection between all the equations mentioned in section 2 and the obtained experimental results. Otherwise, the equations in section 2 are not necessary because they are well known and do not bring anything new to the field;
- Also, I think it would be interesting to be analyzed the topography of the surface layer of the welded samples;
- The resolution of some figures is low, like Figure 5 or Figure 7.
- The discussion part must be much developed in order to be able to highlight the novelty brought by the research presented in the paper in relation to other research in the field. In this form, the novelty of the research compared to previous research cannot be identified. References were made in this section to only 3 bibliographic sources, one of which is from 2008;
- Conclusions should be more concrete and future research directions should be presented.
Author Response
Esteemed reviewer, we would like to express deep gratitude for the attentive reading and thorough analysis of the manuscript. All your comments and remarks are relevant and valuable. While answering your questions, we have found and added a lot of important information. The revised manuscript was expanded due to proper comparison with the relevant literature.
Please find below the answers for all your comments. All the required changes have been made or added to the text. Let us thank you once again for helping us to improve the manuscript.
- It is necessary to make a connection between all the equations mentioned in section 2 and the obtained experimental results. Otherwise, the equations in section 2 are not necessary because they are well known and do not bring anything new to the field;
Esteemed reviewer, thank you for the relevant remark! The following correction has been made in the revised manuscript:
“The optimality area analysis for the USW parameters was carried out according to the approved solving rule (4) over the entire volume under consideration. ANN-simulation results were combined at the summary (collecting) isolines’ graph over the 3D space section. The optimal values of technological parameters were highlighted as the blue-filled regions”.
We have additionally corrected the text with mentioning functional dependence (3), governing the necessity to use the ANN, as well as the solving rule (4), which is responsible for search of the optimal parameters ranges during conducting the experiments.
- Also, I think it would be interesting to be analyzed the topography of the surface layer of the welded samples;
Thank you for the relevant comment. Based on your recommendation, we have added several photographs of fractured test coupons.
- The resolution of some figures is low, like Figure 5 or Figure 7.
You are absolutely correct. The due corrections have been made to the figures.
- The discussion part must be much developed in order to be able to highlight the novelty brought by the research presented in the paper in relation to other research in the field. In this form, the novelty of the research compared to previous research cannot be identified. References were made in this section to only 3 bibliographic sources, one of which is from 2008;
Esteemed reviewer, you are absolutely right. We have substantially enlarged the discussion part with due analysis and citation of the recent publication on the problem. It has really improved the paper.
- Conclusions should be more concrete and future research directions should be presented.
Esteemed reviewer! We completely agree. According to your recommendation, we have corrected the conclusions; future research direction have also been outlined.
Let us thank you once again for many useful pieces of advice and thorough analysis of the manuscripts.
Round 2
Reviewer 1 Report
Thank you for your answers and modifications to the manuscript.
Minor comment: In Table 2, rightmost column, the standard deviation floating point numbers should be formatted consistently (e.g., 0.02, not 0,02).
Author Response
Esteemed reviewer!
Many thanks for accepting our responses. Some addition minor corrections have been made to the text in order to improve the language level.
We highly appreciate your time, valuable comments and advices. It has helped us to improve the text and even our understanding of the problem.
On behalf of the authors,
Sergey Panin
Reviewer 2 Report
The authors revised their manuscript according to my suggestions. Thus the manuscript can be accepted for publication.
Author Response
Esteemed reviewer!
Many thanks for accepting our responses. Some addition minor corrections have been made to the text in order to improve the language level.
We highly appreciate your time, valuable comments and advices. It has helped us to improve the text and even improve the understanding of the problem.
On behalf of the authors,
Sergey Panin